# Effect of High Current Pulsed Electron Beam (HCPEB) on the Organization and Wear Resistance of CeO_2_-Modified Al-20SiC Composites

**DOI:** 10.3390/ma16134656

**Published:** 2023-06-28

**Authors:** Lei Wang, Bo Gao, Yue Sun, Ying Zhang, Liang Hu

**Affiliations:** 1Key Laboratory for Ecological Metallurgy of Multimetallic Mineral (Ministry of Education), Northeastern University, Shenyang 110819, China; surfwangl@163.com (L.W.); surfsuny@163.com (Y.S.); surfzhangy@163.com (Y.Z.); 2Key Laboratory for Anisotropy and Texture of Materials (Ministry of Education), School of Materials Science and Engineering, Northeastern University, Shenyang 110819, China; hss5566123@163.com

**Keywords:** high current pulsed electron beam, Al-20SiC, cerium dioxide, microhardness, wear resistance

## Abstract

This paper investigates the joint effect of high current pulsed electron beam (HCPEB) and denaturant CeO_2_ on improving the microstructure and properties of Al-20SiC composites prepared by powder metallurgy. Grazing Incidence X-ray Diffraction (GIXRD) results indicate the selective orientation of aluminum grains, with Al(111) crystal faces showing selective orientation after HCPEB treatment. Casting defects of powder metallurgy were eliminated by the addition of CeO_2_. Scanning electron microscopy (SEM) results reveal a more uniform distribution of hard points on the surface of HCPEB-treated Al-20SiC-0.3CeO_2_ composites. Microhardness and wear resistance of the Al-20SiC-0.3CeO_2_ composites were better than those of the Al matrix without CeO_2_ addition at the same number of pulses. Sliding friction tests indicate that the improvement of wear resistance is attributed to the uniform dispersion of hard points and the improvement of microstructure on the surface of the matrix after HCPEB irradiation. Overall, this study demonstrates the potential of HCPEB and CeO_2_ to enhance the performance of Al-20SiC composites.

## 1. Introduction

With the development of industrialization, the performance of wear-resistant materials has put forward higher requirements. Composite materials have emerged as an indispensable and important part of the wear-resistant field, such as SiC particle-reinforced aluminum matrix composite with high hardness and good wear resistance. It is a comprehensive composite of excellent friction materials with promising applications [1,2,3,4,5,6,7]. It has been used to some extent in engine pistons and cylinder liners in high-speed train brake discs, automobile brake discs, and sliding bearing materials. At present, the conventional methods for preparing Al-20SiC composites include the extrusion casting method, in situ reaction method, stirring casting method, spray deposition method, powder metallurgy method, and so on [8]. Among them, the powder metallurgy method has the advantages of easy control of the interfacial reaction, low preparation cost, and significantly better performance and stability of the material than those prepared by other methods [9]. However, this preparation method produces significant casting defects, such as porosity, cracking, and agglomeration, which limit the wide application of Al-20SiC composites [10]. Therefore, it is necessary to improve the properties of Al-20SiC composites using different treatment methods [11].

High current pulsed electron beam (HCPEB) surface treatment technology is an emerging high-energy beam surface treatment technology with higher energy efficiency than laser beams and is not affected by ionic impurities in ion beam technology. With high energy density and short pulse electron beam irradiation of the material surface, the electron beam carries the energy deposited to the material surface, resulting in instantaneous heating and cooling and the formation of a special surface microstructure. Then, the expected corrosion resistance, wear resistance, and other surface performance effects are obtained [12]. Walker et al. [13] treated eutectic Al-Si alloy using a pulsed electron beam and showed that the average surface roughness increased and then decreased with increasing accelerating voltage; in addition, the average dynamic friction coefficient of the treated sample was higher than that of the untreated alloy surface and increased the wear rate by 66%. Hao et al. [14] used HCPEB for the surface treatment of a peri-eutectic Al-15Si alloy, and the results of the study showed that the alloy elements Al and Si undergo mutual diffusion in the molten state of the electron beam accompanied by a transient solidification effect, and a supersaturated solid solution of aluminum is formed in the surface layer of the alloy with enhanced overall wear resistance. At the same time, some literature studies have shown that the addition of trace amounts of metamorphic agents can improve the irradiation damage of metals during HCPEB irradiation and improve the overall properties of the matrix. Shi Weixi et al. [15] added 0.3 wt.% Nd to Al-20Si alloy material, and the results show that the grain size was significantly reduced and the wear resistance was significantly improved. Hu et al. [16] studied the elimination of microcracks on the surface of Al alloy samples by rare earth elements, and the results showed that the addition of trace rare earth elements reduced the density of microcracks to some extent, which led to grain refinement and enhanced the alloy’s resistance in a corrosion solution [17].

Among rare earth elements, CeO_2_ is an important denaturing agent that can form a uniform distribution in aluminum matrix composites, effectively improving the strength and hardness of the material. Additionally, CeO_2_ can enhance the material’s crystal structure and overall performance [18,19]. Compared to other denaturing agents, CeO_2_ is relatively inexpensive, which can reduce the production cost of the material and improve its economic competitiveness. In this study, the utilization of the HCPEB surface modification technique aims to address the inherent defects commonly found in powder metallurgy. Additionally, the incorporation of rare earth oxide CeO_2_ as a trace additive is expected to enhance the elimination of microcracks and pores within the material. Notably, this research represents the first investigation into the wear resistance of HCPEB-modified pre- and post-Al-20SiC composites, providing a comprehensive understanding of the underlying mechanisms governing wear resistance [20,21].

## 2. Materials and Methods

### 2.1. Material Preparation

The raw materials used in this experiment were commercially available aluminum powder, SiC powder, cerium oxide powder, and hydroxypropyl methylcellulose (HPMC). The raw material composition and particle sizes are shown in Table 1.

The specific preparation process is as follows: The raw materials—aluminum powder, SiC powder, cerium oxide powder, and 1 wt.% hydroxypropyl methyl cellulose (HPMC)—were put into the ball mill tank. At the same time, zirconium oxide agate balls were added with a ratio of 3:1 and mixed on a roller mixer at 300 r/min for 5 h. A total of 3 g of well-mixed material was weighed on an electronic balance and loaded into a mold with specifications of Φ25 × 100 mm and 5 mm × 5 mm × 50 mm to press the raw material into a molding system. The billets were mechanically compacted in a cold isostatic press and dried in a vacuum drying oven under vacuum at 80 °C for 3 h. Finally, the billets were sintered in a tubular resistance furnace at a heating rate of 9 °C/min for 8 h at 590 °C to produce Al-20SiC-0.3CeO_2_ composites. Prior to HCPEB treatment, the material was cut into 10 mm × 10 mm × 5 mm samples by a metal cutter and then polished sequentially using sandpaper (100#, 240#, 400#, 800#, 1500#, 3000#) and diamond polishing paste (~1 µm).

### 2.2. HCPEB Treatment

The surface modification of the material was carried out by the HOPE-I type HCPEB treatment device manufactured by the Dalian University of Technology (Liaoning, China). The relevant process parameters are shown in Table 2. The number of pulses for each experiment was 5, 15, and 25.

### 2.3. Microstructure Characterization and Performance Analysis

Grazing Incidence X-ray Diffraction was performed using a multifunctional X-ray diffractometer (model X’Pert PRO, Panaco, The Netherlands). The friction coefficient of the aluminum matrix composite surface before and after the electron beam modification was measured by the friction test using the reciprocating motion mode on the surface of the sample with the wear instrument “MTF-5000” (Atech Instruments Technology Co., Ltd., Nanjing, China). The friction conditions were as follows: Si_3_N_4_ ball, 2 N load, 7 mm friction distance, and 10 min friction test time, and a Hitachi S-4800 field emission scanning electron microscope (Hitachi High-Tech Corporation, Tokyo, Japan) was used to observe the wear morphology on the surface of aluminum matrix composites with and without the addition of rare earth oxide (CeO_2_), before and after the HCPEB modification. For the measurement of the surface microhardness of Al-20SiC composites before and after the HCPEB treatment, a Vickers hardness tester of type LM247AT (Luotai Precision Instruments Co., Ltd., Dongguan, China.) was selected. The test parameters were: load of 200 g and holding time of 13 s.

## 3. Results

Figure 1 illustrates the mixing of the raw material powders after ball milling. From the figure, it can be seen that the Al powder, SiC powder and CeO_2_ powder are uniformly mixed. After ball milling, each powder is broken and refined to some extent.

Figure 2 shows the Grazing Incidence X-ray Diffraction (GIXRD) patterns of the Al-20SiC-0.3CeO_2_ composite samples before and after the electron beam treatment. Figure 2 shows that the electron beam treatment of the sample with the addition of rare-earth Ce did not result in the formation of new phases. The sample mainly consisted of two phases, Al and SiC, and no rare-earth Ce-rich phases were detected. This is likely due to the fact that the content of rare-earth Ce in the alloy is very small and falls below the detection limit of the GIXRD instrument. The HCPEB treatment induced rapid melting and solidification processes that altered the original oriented casting organization in the surface layer of the alloy [22]. The temperature and stress fields induced by the electron beam induce severe plastic deformation on the surface of the alloy because of the FCC structure of aluminum and the high number of slip systems. Yan et al. [23] found a significant enhancement in the intensity of the diffraction peak of Al(220) in the surface-modified layer of 2024-type aluminum alloy treated with HCPEB, showing a selective growth of Al(110) grains in the modified layer. Hao et al. [24] treated AISI 316L stainless steel with HCPEB and found plastic deformation during the modification process, with a selective orientation of the (111) grain surface. It can be seen from Figure 2 that the crystal orientation of the surface layer of the sample was changed after five pulse treatments, and the tendency of aluminum grains to grow along the Al(111) and Al(200) crystal planes in a meritocratic manner was enhanced. Since the Al(111) crystal faces are tightly packed with minimal surface energy and good stability, it is beneficial to improve the substrate microstructure and wear resistance [25]. After 25 pulse treatments, the diffraction peak of Al(111) was shifted to a high angle, attributed to the generation of residual compressive stress. Residual compressive stresses can improve the wear resistance of materials by reducing the likelihood of surface damage and wear [26].

Figure 3 displays the surface microstructure morphology of the Al-20SiC composites before and after HCPEB treatment. The original histomorphology showed that the gray silicon carbide phase was uniformly distributed in the aluminum matrix, and the size of the SiC particles was around 10 μm (Figure 3a). The vicinity of the SiC particles exhibited a number of pore structures, which is believed to be due to the high viscosity of the liquid Al phase at the low sintering temperature of 590 °C [27]. This led to relatively poor mobility of the aluminum liquid and prevented the material from completing the complementary shrinkage during the subsequent solidification process, eventually resulting in the pore structure [28]. As shown in Figure 3b–d, an increase in the number of pulses results in the evaporation of SiC particles from the subsurface and their eventual eruption from the melting surface, forming a characteristic crater morphology. The EDS results (Figure 3e) indicate that the particles erupted at A are likely composed of SiC or Si. Previous studies have demonstrated that microstructural irregularities, such as grain boundaries, phase boundaries, and second-phase particles, are more prone to serve as nucleation centers for the formation of these crater-shaped features [29].

Figure 4 depicts the surface microstructure morphology of Al-20SiC-0.3CeO_2_ composites before and after HCPEB treatment. Compared to the sample without the addition of rare earth oxides in Figure 3a, there are fewer pores present in the original sample after the addition of CeO_2_, as seen in Figure 4a [30]. This is mainly due to the fact that the addition of CeO_2_ can significantly enhance the wettability between the liquid phase Al and the solid phase SiC, resulting in relatively dense samples during the sintering process. The mechanism of crater formation in Figure 4c,d is not explained as previously mentioned. Figure 4e shows that the surface of the aluminum matrix is covered with an oxide film. The oxide layer improves the wear resistance of the aluminum matrix composite surface because the oxide layer has a high hardness and resists friction and wear. In addition, the oxide layer provides lubrication to the material surface and reduces the coefficient of friction. From Figure 4f, it can be concluded that the rare earth CeO_2_ is able to diffuse during the HCPEB treatment due to the electron beam’s role in promoting the elemental diffusion effect [31]. In addition, CeO_2_ reacts with impurities in the remelted layer in the HCPEB action zone and weakens the local stress concentration in the brittle phase melt pool. It also reduces the surface tension, leading to a smaller contact angle between Al and SiC, which is conducive to the elimination of pores and greatly improves the surface microstructure [2].

Figure 5 shows the changes in the aluminum-based Vickers hardness of Al-20SiC and Al-20SiC-0.3CeO_2_ alloys before and after the intense current pulsed electron beam treatment. It can be observed from the figure that the Vickers microhardness of both Al-20SiC and Al-20SiC-0.3CeO_2_ is greatly improved after the electron beam treatment, and the hardness tends to increase with the increase in the number of pulses [32]. The average value of Al matrix microhardness before HCPEB treatment was 50.2 HV, the average value of Al matrix microhardness for 5 pulses was 71.4 HV, the average value of Al matrix microhardness for 15 pulses was 109.5 HV, and the value of Al matrix microhardness for Al-20SiC-0.3CeO_2_ composites after 25 pulses was the largest with an average value of 130.1 HV [33]. As seen in Figure 4, the HCPEB treatment of the material with the addition of the rare earth oxide CeO_2_ increases the uniformity of the hard-phase particle distribution, which enhances the load-bearing capacity of the matrix aluminum per unit area. At the same time, the addition of CeO_2_ improves the mobility of alloying elements and reduces tissue sparseness, resulting in better bonding of the reinforcing phase and the Al matrix, increasing the stress tolerance of the Al matrix, and increasing the hardness of the material. Ahmad et al. [34] showed that when the surface of the alloy was treated with an electron beam, the reinforcing phase SiC was dissolved and broken into fine particles uniformly distributed in the Al matrix under the action of the electron beam. The hardness of the primary phase is increased due to the uniform dispersion of the hard points, and the hardness of the coating is significantly enhanced due to the large amount of hard-phase SiC in the coating, which acts as a barrier to dislocation movement. CeO_2_ is mostly located at grain boundaries or phase boundaries, which significantly reduces the activity of the interface and hinders the diffusive movement of aluminum grain boundaries, which can play a certain role in nailing the aluminum grain boundaries, causing the deformation of the matrix aluminum to be hindered, increasing the deformation resistance, and increasing the hardness [2]. The microhardness of the material surface is increased, thus enhancing to some extent the bearing effect of the composite material on stresses during frictional wear while causing less plastic deformation on the material surface during frictional wear and reducing the friction coefficient [35].

Figure 6 demonstrates the evolution of the friction coefficient with friction time and the corresponding friction coefficient of the Al-20SiC-0.3CeO_2_ composite surface for different numbers of pulses [36]. From Figure 6a, it can be seen that the friction profile of the untreated and Al-20SiC-0.3CeO_2_ composite surface under five pulses fluctuates more, and the width and depth of the profile also vary visually [37]. However, the friction curves of the Al-20SiC-0.3CeO_2_ composite surface after 15 pulses and 25 pulses steadily floated, with small width and shallow depth of the profile. From Figure 6b, it can be seen that the friction coefficient on the surface of Al-20SiC-0.3CeO_2_ composites shows an overall trend of decreasing all the time with the increase in the number of pulses. The friction coefficient of the specimen without HCPEB treatment is 0.520, and the friction coefficient on the surface of the specimen reaches the minimum value of 0.111 at 25 pulses, with a decrease of 78.65%. Shi Weixi [15] et al. added rare-earth Nd to an Al-Si alloy, and the results showed that the addition of rare earths resulted in a significant refinement of the primary silicon organization located in the alloy and a stronger bond with the matrix; thus, fatigue damage and chipping were greatly reduced, which in turn greatly improved the wear resistance of the material. The specific mechanism of the action of the composite surface wear resistance is further described below.

Figure 7 illustrates the wear rate changes of Al-20SiC and Al-20SiC-0.3CeO_2_ before and after HCPEB irradiation. As shown in the graph, the wear rate of the sample with CeO_2_ addition is lower than that of the sample without CeO_2_ at the same number of pulses; the wear rates of the samples decreased from 8.84 and 6.17 to 2.89 and 1.93, respectively, as the pulse number increased. Combined with the friction coefficient curve, the results indicate that the wear resistance of the samples is best after the 25-pulse treatment. The addition of CeO_2_ contributes to the improvement of wear resistance.

Figure 8 shows the wear morphology of the surface of Al-20SiC composites before and after 25-pulse treatments. During the frictional wear of the aluminum matrix composite, the SiC hard particles—as the main load-bearing phase—are subjected to both positive and tangential stresses [38]. From the wear morphology, it can be seen that on the untreated Al-20SiC composite, the wear surface showed a debris-like flaking phenomenon and a certain degree of plastic deformation occurred along the sliding direction on both sides of the plow groove. The stripped SiC constituted abrasive wear during the friction process or entered between the friction subsets, forming scratches and grooves on the wear surface [39]. At the same time, abrasive wear was further accelerated by the generation of hard particles and adherence to the surface during the sliding wear test, which produced microcuttings in the surface layer. The worn surface showed deep grooves in addition to furrow abrasions, and the wear mechanism was abrasive wear [40]. The wear surface of the samples after 25-pulse treatments was flatter, relatively smooth, and with shallow scratches, and the wear rate after modification was lower than the wear rate before modification.

Figure 9 shows the morphology of the worn surface of Al-20SiC-0.3CeO_2_ composites before and after 25-pulse treatments. It can be seen from the figure [41] that the untreated Al-20SiC-0.3CeO_2_ composite has obvious plow-like stripes of different widths and depths on the wear surface, and some of the wear surfaces have traces of being cut. Additionally, relatively deep grooves appear, which indicates that the matrix alloy wears relatively severely, and its wear mechanism is typical of adhesive wear [42]. This is due to the low hardness of the matrix alloy material and the tearing wear caused by the plowing action of the anti-abrasive during frictional wear. For the Al-20SiC-0.3CeO_2_ composite treated with 25 pulses, the wear surface is relatively flat with no obvious groove-like streaks. Micron-sized particles can be seen on the matrix that were ground off and not dislodged but flatly exposed to the wear surface. This indicates that the regrind particles are directly subjected to frictional wear and play the main load-bearing role, and their wear mechanism is typical of abrasive wear [43]. It can be seen from the figure that the addition of rare earth oxides gives a lower wear rate at the same number of pulses and improves the wear resistance of the material surface.

The HCPEB treatment plays a pivotal role in optimizing the microstructure of Al-20SiC composites. The intense current pulses generated by HCPEB initiate rapid melting and solidification processes, resulting in improved grain orientation. This optimized microstructure significantly enhances the material’s resistance to wear by minimizing crack propagation, reducing surface deformation, and mitigating fatigue. Additionally, CeO_2_ strengthens the bonding between the reinforcing phase (SiC) and the aluminum matrix, effectively reducing the occurrence of defects such as pores and microcracks. This improved bonding enhances the overall integrity and strength of the composite material, resulting in enhanced wear resistance [44]. Finally, CeO_2_ enhances the mobility of alloying elements within the material. This increased mobility allows for better dispersion of hard-phase particles, such as SiC, throughout the aluminum matrix. The uniform dispersion of these hard-phase particles leads to heightened hardness and improved resistance to wear and abrasion.

## 4. Results and Discussion

### 4.1. Results

In this paper, the effect of CeO_2_ on the properties of Al-20SiC composites after HCPEB treatment was investigated.

(1)The results of GIXRD analysis showed that the rapid melting and solidification processes triggered by the HCPEB treatment led to selective orientation of the matrix and selective growth of Al(111) grains in the modified layer.(2)SEM results showed that the presence of rare earth elements effectively eliminated defects such as porosity.(3)Hardness tests showed that the addition of rare earth Ce increased the average microhardness of the matrix by 159.16%.(4)The friction coefficient showed a reduction of 87.18% with the synergistic effect of CeO_2_ and HCPEB.

### 4.2. Discussion

The optimal growth of the Al(111) crystal plane improves the material’s crystal structure and grain orientation, benefiting from its favorable crystallization properties and high-density arrangement. This well-ordered arrangement of grains effectively withstands external stress and wear, resulting in reduced surface wear and fatigue of the material.

The increase in surface hardness can be attributed to two factors: Firstly, the addition of CeO_2_ enhances the activity of alloying elements, reduces tissue sparseness, and improves the bonding between the reinforcing phase and the aluminum matrix. Secondly, the treatment of the alloy surface with an electron beam results in the dissolution and fragmentation of the reinforcing phase SiC into fine particles that are uniformly dispersed within the aluminum matrix under the beam’s action.

The improvement in wear resistance is mainly attributed to the enhanced hardness of the material’s surface, which enhances its resistance to scratches and wear. Additionally, the HCPEB treatment induces rapid melting and solidification, optimizing the microstructure of the material and further contributing to its enhanced wear resistance.

## Figures and Tables

**Figure 1 materials-16-04656-f001:**
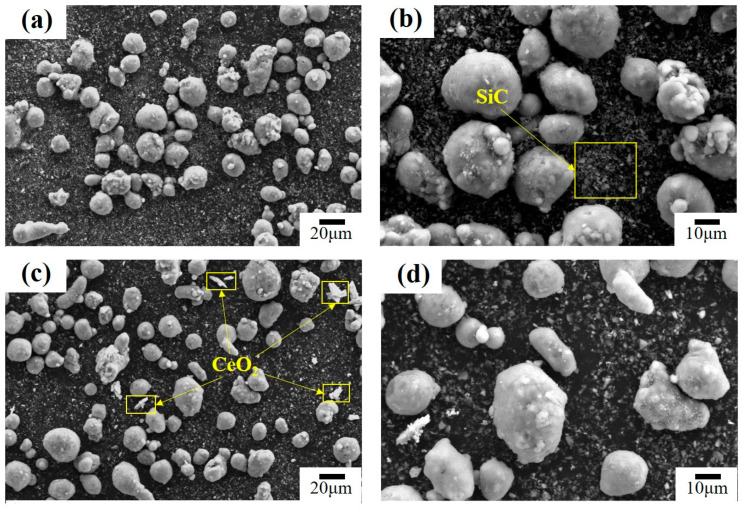
Mixing powder after ball milling: (**a**) Al-20SiC mixing powder, (**b**) partial enlargement, (**c**) Al-20SiC-0.3CeO_2_ mixing powder, and (**d**) partial enlargement.

**Figure 2 materials-16-04656-f002:**
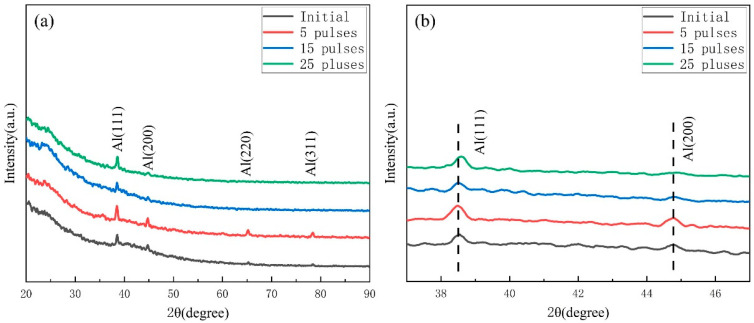
GIXRD patterns of Al-20SiC-0.3CeO_2_ samples before and after HCPEB irradiation: (**a**) complete GIXRD patterns; (**b**) enlarged patterns of Al(111) crystal plane and Al(200).

**Figure 3 materials-16-04656-f003:**
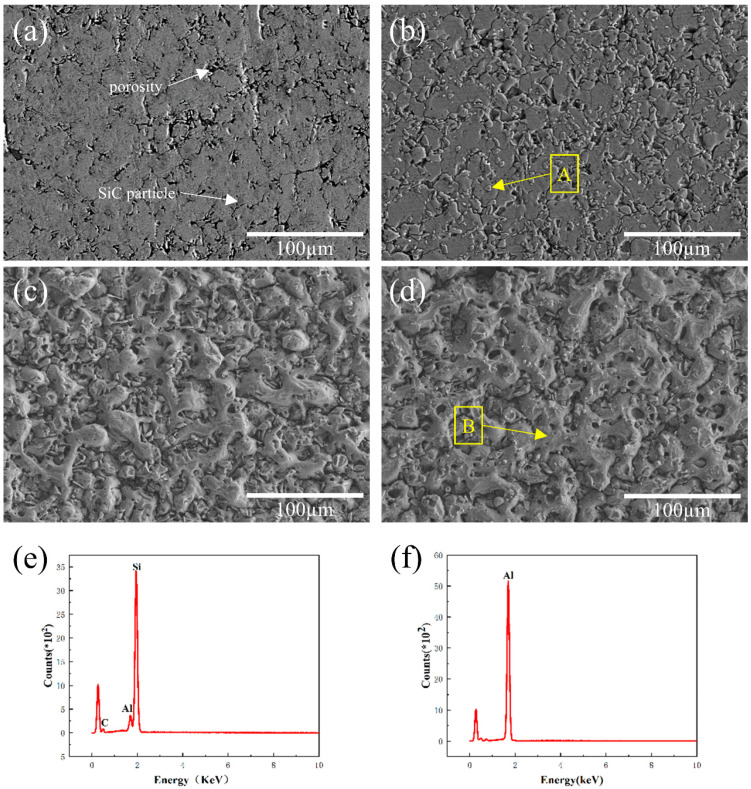
Surface morphology of Al-20SiC sample before and after strong current pulsed electron beam treatment. (**a**) Original sample; (**b**) 5 pulses; (**c**) 15 pulses; (**d**) 25 pulses; (**e**) EDS results of the A region in Figure 3b; (**f**) EDS results of the B region in Figure 3d.

**Figure 4 materials-16-04656-f004:**
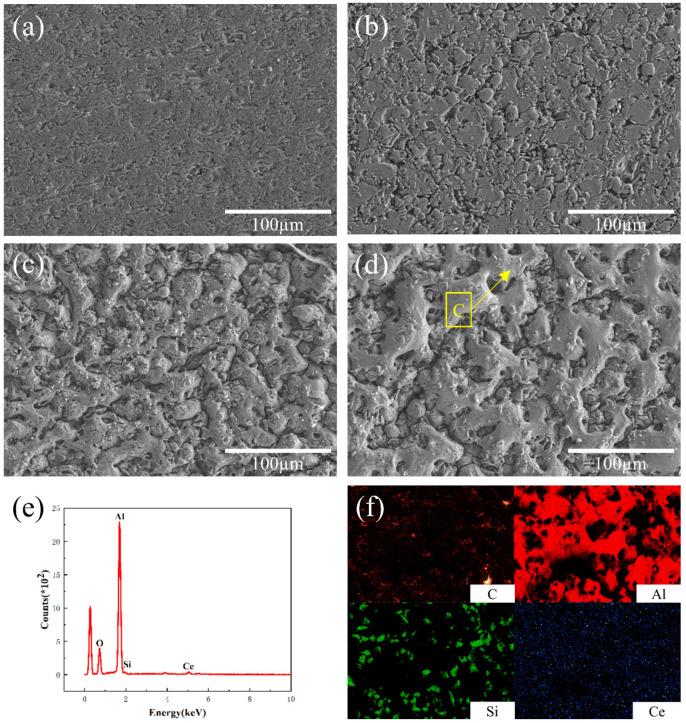
Surface morphology of Al-20SiC-CeO_2_ sample before and after strong current pulsed electron beam treatment. (**a**) Original sample; (**b**) 5 pulses; (**c**) 15 pulses; (**d**) 25 pulses; (**e**) EDS results of the C region in Figure 4d; (**f**) alloy surface element distribution at 25 pulses.

**Figure 5 materials-16-04656-f005:**
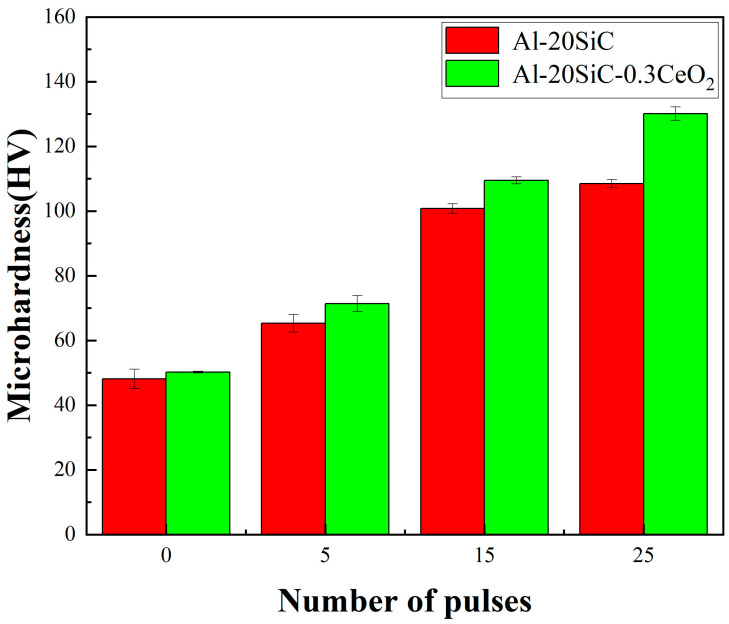
Microhardness measurements of Al-20SiC and Al-20SiC-0.3CeO_2_ samples under different pulse times.

**Figure 6 materials-16-04656-f006:**
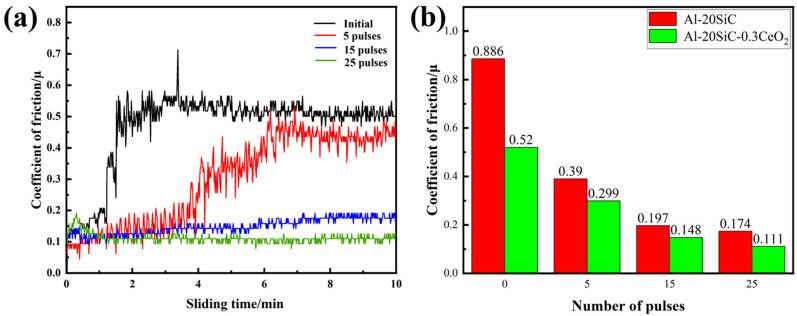
(**a**) Variation curves of friction coefficient of Al-20SiC-0.3CeO_2_ with friction time for different pulse numbers; (**b**) surface friction coefficients of Al-20SiC and Al-20SiC-0.3CeO_2_ for different pulse numbers.

**Figure 7 materials-16-04656-f007:**
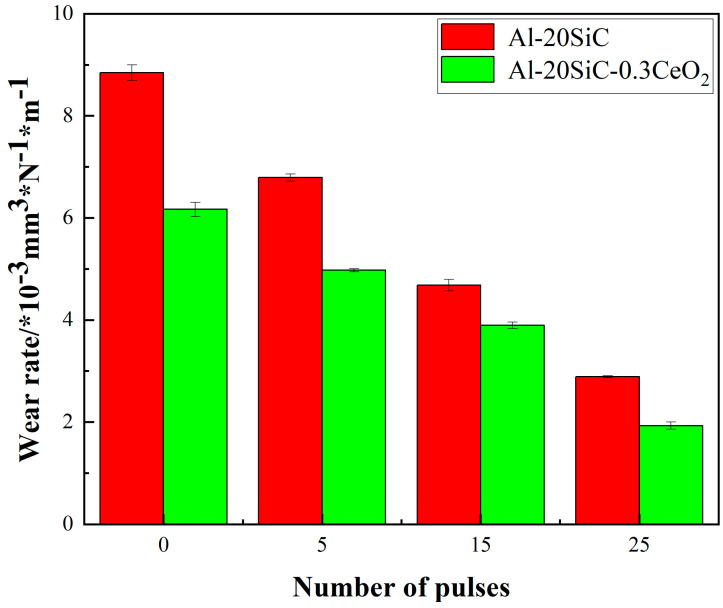
Variation of sample wear rate before and after HCPEB irradiation.

**Figure 8 materials-16-04656-f008:**
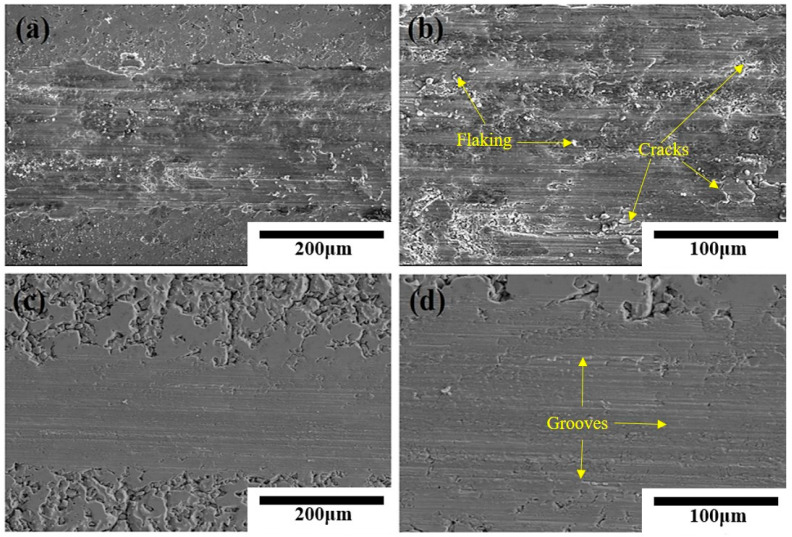
Surface wear morphology of Al-20SiC before and after electron beam treatment. (**a**) Original sample; (**b**) local enlarged image of original sample; (**c**) 25 pulses; (**d**) local enlarged image of 25 pulses.

**Figure 9 materials-16-04656-f009:**
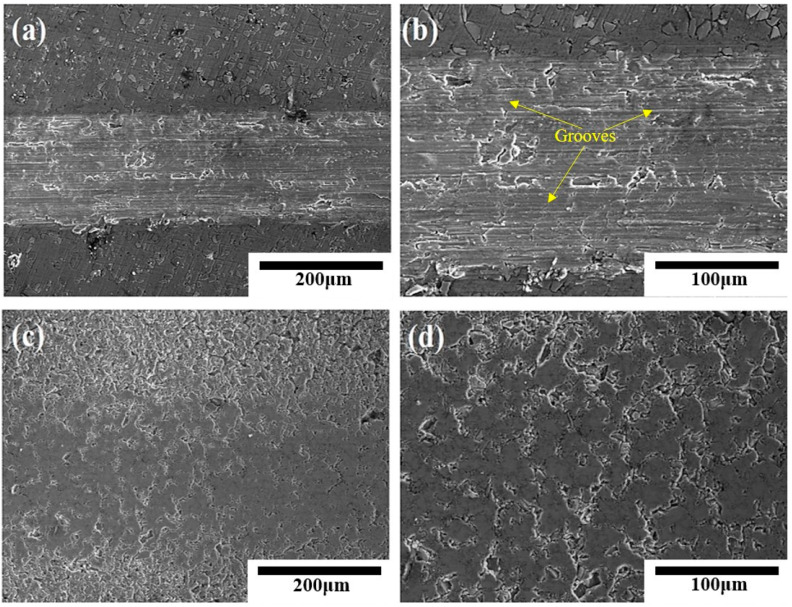
Surface wear morphology of Al-20SiC-0.3CeO_2_ before and after electron beam treatment. (**a**) Original sample; (**b**) local enlarged image of original sample; (**c**) 25 pulses; (**d**) local enlarged image of 25 pulses.

**Table 1 materials-16-04656-t001:** Composition and granularity of raw materials.

Powder	Purity/wt.%	Particle Size/µm
Al	99.9 wt.%	20–30 µm
SiC	99.9 wt.%	1–2 µm
CeO_2_	99.9 wt.%	6–10 µm

**Table 2 materials-16-04656-t002:** Working parameters of the HCPEB system.

Acceleration Voltage (kV)	Energy Density (J/cm^2^)	Pulse Time (μs)	Pulse Frequency (Hz)	Vacuum Level (Pa)
24.5	2	3	0.1	6.5 × 10^−3^

## Data Availability

Data will be made available upon request from the corresponding author.

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
