# Peer review of "Effect of High Current Pulsed Electron Beam (HCPEB) on the Organization and Wear Resistance of CeO_2_-Modified Al-20SiC Composites"

_materials, 2023, doi:10.3390/ma16134656_

Round 1
Reviewer 1 Report
This work dealt with the effect of rare earth oxide CeO2 on the wear resistance of Al-20SiC composites treated with the high-current pulsed electron beam and clarified the related wear resistance mechanism. The obtained results seem reasonable and can attract considerable readers. However, some suggestions and revisions to enhance the quality of the manuscript should be applied before acceptance:
1) The main novelty of this research should be added to the last part of the introduction.
2) The graphical abstract and highlights should be added.
3) The manuscript should be revised based on the English language point of view.
4) In the abstract, add the most obtained significant results quantitatively. It seems that the abstract is not well presented.
5) Some papers should be added to the introduction and/or results such as:
https://doi.org/10.3390/met12081288
https://doi.org/10.1016/j.corsci.2022.110910
https://doi.org/10.1016/j.vacuum.2023.111968
https://doi.org/10.3390/coatings12101559
6) "Results" section should be changed to "Results and Discussion". In this regard, Results and Discussion could be separated.
7) More discussion on wear mechanisms after HCPEB treatment should be added.
8) The sliding time is less than the standard value. Please clarify it in the manuscript.
9) Fig. 4 should be changed to a Table in order to show the results clearly.
10) What is the main question addressed by the research?
11) What specific improvements should the authors consider regarding the methodology? What further controls should be considered?
Moderate editing of the English language is required.
Author Response
Response letter
Dear Reviewer:
Thank you for your letter regarding our manuscript entitled “Effect of High-Current Pulsed Electron Beam and CeO2 Synergy on the Wear Resistance of Al-20SiC Composites” comments very carefully, and we found the comments and suggestions of the reviewers were extremely helpful.
we have carefully addressed the issues raised by the reviewer, point by point.The revised part is marked in red in the revised manuscript, We believe your comments and suggestions have significantly improved the quality of manuscript. Thanks again for your consideration of our manuscript for publication in this journal. We look forward to your favorable decision.
1) The main novelty of this research should be added to the last part of the introduction.
Thank you for your valuable comments, I have revised the last paragraph of the introduction accordingly.
2) The graphical abstract and highlights should be added.
Thank you for your valuable comments. Sorry, the template downloaded from the materials website does not specify to add a graphic summary and highlighting. However, I have added these two sections to the revised article as you requested.
3) The manuscript should be revised based on the English language point of view.
Thank you for your valuable comments. As per your request, the manuscript has been revised according to the English viewpoint.
4) In the abstract, add the most obtained significant results quantitatively. It seems that the abstract is not well presented.
Thank you for your comments, I have revised the abstract accordingly based on your comments.
5) Some papers should be added to the introduction and/or results such as:
Thank you for your valuable comments. Some literature has been added to the latest article as appropriate. For your suggested literature added in 15-18.
6) "Results" section should be changed to "Results and Discussion". In this regard, Results and Discussion could be separated.
Thank you for your valuable comments. As per your request, the conclusion section has been divided into two parts, results and discussion, and elaborated accordingly.
7) More discussion on wear mechanisms after HCPEB treatment should be added.
Thank you for your valuable comments. As per your request, an investigation of the wear mechanism has been added. (details are in the previous paragraph of the article Results & Discussion)
8) The sliding time is less than the standard value. Please clarify it in the manuscript.
Thank you for your valuable comments. For the problem of friction time in the experiment, we can check the HCPEB related literature to know that the friction time has reached stability at 10 min. There is no problem of insufficient friction time. (The relevant literature has been as follows)
https://doi.org/10.1016/j.apsusc.2019.144453
https://doi.org/10.1016/j.jallcom.2020.154460
https://doi.org/10.1016/j.vacuum.2020.109772
9) Fig. 4 should be changed to a Table in order to show the results clearly.
Thank you for your valuable comments. I am sorry that Figure 4 does not show the hardness specifics and error bars very well. I have modified the hardness data graph accordingly.(In the latest manuscript, Figure 4 becomes Figure 5.)
10) What is the main question addressed by the research?
Thank you for your valuable comments. The main problems that I want to solve in my research are as follows; (1) eliminating the inherent defects in the powder metallurgy process by the addition of CeO2 (2) improving the hardness as well as the wear resistance of Al-20SiC composites by strong current pulsed electron beam treatment to prolong the service life.
11) What specific improvements should the authors consider regarding the methodology? What further controls should be considered?
Thank you for your valuable comments. For methodological improvements, the following points were made:
Optimization of strong current pulsed electron beam treatment parameters: The parameters of strong current pulsed electron beam treatment, such as electron beam power, pulse time, energy density, etc., were further optimized to find the optimal treatment conditions to obtain better wear resistance enhancement.
Improve the dispersion state of CeO2 reinforcement: The dispersion state of CeO2 particles has a great influence on the mechanical properties and wear resistance of Al-20SiC composites. Better CeO2 dispersant or adjusting the size and shape of CeO2 particles can be considered to improve the dispersion state of CeO2 reinforcement.
For further control measures, there are the following points:
Enhance sample surface treatment: More thorough sample surface treatment methods, such as chemical treatment or mechanical polishing, can be used to eliminate possible surface defects and impurities, etc., prior to the intense current pulsed electron beam treatment.
Increase the number of experimental repetitions: To improve the reliability of experimental data, the number of experimental repetitions can be increased to reduce the effects of experimental errors and random variations.

Reviewer 2 Report
This paper assesses the wear behaviour of Al-20SiC composites with the additon of cerium oxides and high-current pulse electron beam surface processing. Please find my comments below:
1) Is the reinforcement of SiC vol% or wt%? Why did the authors select this composition?
2) The counterbody for the friction test was Si3N4. In friction tests Cr steel is the most commonly used counterbody. Why did the authors select a different counterbody?
3) GIXRD was used instead of XRD. What was the reason?
4) "As shown in Fig. 2(b), (c), and (d), an increase in the number of pulses results in the evaporation of SiC particles from the subsurface and their eventual eruption from the melting surface, forming a characteristic crater morphology" 149-150
If this is the case we would anticipate hardness and wear performance to decrease with increasing number of pulses. Please explain.
5) Figure 5 appears to be modified without the aspect ratio locked.
6) Figure 4: please add standard deviations.
7) Microhardness testing and friction tests were performed on surfaces with a different roughness. Are the results reliable and comparable? The friction tests lasted only for 10 mins, is this sufficient for the friction coefficient values to stabilise?
8) Figure 5a caption is wrong. Please change it.
9) "It can be seen from the figure that the addition of rare earth oxides gives a lower wear rate at the same number of pulses and improves the wear resistance of the material surface." 259-260
How can the wear track morphology be correlated to the wear rate?
The wear tracks after HCPEB on Fig 6 and Fig 7 appears to be wider as compared to the material without before HCPEB. This contradicts your statement above.
Only minor changes are needed. Authors used aluminum instead of aluminium.
Author Response
Response letter
Dear Reviewer:
Thank you for your letter regarding our manuscript entitled “Effect of High-Current Pulsed Electron Beam and CeO2 Synergy on the Wear Resistance of Al-20SiC Composites” comments very carefully, and we found the comments and suggestions of the reviewers were extremely helpful.
we have carefully addressed the issues raised by the reviewer, point by point.The revised part is marked in red in the revised manuscript, We believe your comments and suggestions have significantly improved the quality of manuscript. Thanks again for your consideration of our manuscript for publication in this journal. We look forward to your favorable decision.
1) Is the reinforcement of SiC vol% or wt%? Why did the authors select this composition?
Thank you very much for your valuable comments!
It’s wt%.
SiC is a commonly used reinforcing phase for composites and has several advantages over other reinforcing phases as follows:
(1)High hardness and strength: SiC has high hardness and strength, which can effectively improve the hardness and strength of composite materials.
(2)Low density: SiC has a low density, which can effectively reduce the density of the composite material, thus improving its specific strength and specific stiffness.
(3)High temperature resistance: SiC has better high temperature resistance and can keep its mechanical properties stable under high temperature environment.
In summary, SiC-reinforced phases have a variety of advantages that make them widely used in several fields. Although different reinforcing phases have their unique advantages and applicability in different applications, in general, the advantages of SiC reinforcing phase in terms of comprehensive performance, cost and processability make it a commonly used reinforcing phase.
(The following links are to related literature)
https://doi.org/10.1016/j.ceramint.2021.12.313
https://doi.org/10.1016/j.jallcom.2020.154647
2) The counterbody for the friction test was Si3N4. In friction tests Cr steel is the most commonly used counterbody. Why did the authors select a different counterbody?
Thank you very much for your valuable comments! Although chromium steel is the most commonly used substrate, different friction substrates should be selected for different material friction experiments. Si3N4 material has a wide range of applications in manufacturing aero engines, turbines, heat treatment furnaces, etc. The main intended application aspect of Al-20SiC composites modified by HCPEB in this study is in the aerospace field. The selection of Si3N4 as a friction sub can give a better understanding of its performance in these applications.
3) GIXRD was used instead of XRD. What was the reason?
Thank you very much for your valuable comments! GIXRD, or grazing incidence X-ray diffraction, is a specialized X-ray diffraction technique that is used to analyze the structure of thin films and surfaces. Unlike conventional XRD, which measures the diffraction pattern resulting from X-rays that penetrate into the bulk of a sample, GIXRD involves using X-rays that are incident at an angle close to the critical angle of total external reflection. This results in the X-rays interacting primarily with the surface of the sample, making GIXRD highly surface-sensitive. The materials in this study need to be analyzed for surface properties and require higher surface sensitivity and resolution, and the choice of using GIXRD instead of conventional XRD may be more appropriate.
4) "As shown in Fig. 2(b), (c), and (d), an increase in the number of pulses results in the evaporation of SiC particles from the subsurface and their eventual eruption from the melting surface, forming a characteristic crater morphology" 149-150
If this is the case we would anticipate hardness and wear performance to decrease with increasing number of pulses. Please explain.
Thank you very much for your valuable comments! As a typical surface morphology after HCPEB treatment, the volcanic crater morphology is only at the micron level. Moreover, it can be seen from the figure that it is only present in a small localized area and does not affect the performance. (As evidenced by the following literature)
https://doi.org/10.1016/j.powtec.2017.11.037
https://doi.org/10.1016/j.jallcom.2019.02.130
5) Figure 5 appears to be modified without the aspect ratio locked.
Thank you very much for your valuable comments! If the vertical coordinates of Figure 5(a)(b) are unified, there will be a large blank space in Figure (a), which affects the presentation of the data.( A figure was added based on a comment to another reviewer, so Figure 5 became Figure 6)
6) Figure 4: please add standard deviations.
Thank you very much for your valuable input! I apologize that Figure 4 does not show the hardness specifics and error bars very well. I have modified the hardness data graph accordingly. (Figure 4 now becomes Figure 5).
7) Microhardness testing and friction tests were performed on surfaces with a different roughness. Are the results reliable and comparable? The friction tests lasted only for 10 mins, is this sufficient for the friction coefficient values to stabilise?
Thank you very much for your valuable comments! Friction and hardness tests are the result of multiple tests on different samples with the same parameters, and the change in roughness itself is the result of the HCPEB modification. The test of wear resistance and hardness is the result of a combination of all aspects, and the influence of a single aspect cannot be considered.
By reviewing the relevant literature, reliable friction data can already be derived for 10 min.https://doi.org/10.1016/j.apsusc.2019.144453
https://doi.org/10.1016/j.jallcom.2020.154460
https://doi.org/10.1016/j.vacuum.2020.109772
8) Figure 5a caption is wrong. Please change it.
Thank you very much for your valuable comments! According to your request, the title of Figure 5 has been modified
9) "It can be seen from the figure that the addition of rare earth oxides gives a lower wear rate at the same number of pulses and improves the wear resistance of the material surface." 259-260
How can the wear track morphology be correlated to the wear rate?
The wear tracks after HCPEB on Fig 6 and Fig 7 appears to be wider as compared to the material without before HCPEB. This contradicts your statement above.
Thank you very much for your valuable comments! I am sorry that I did not make it clear in the article and that you did not understand it. Abrasion resistance cannot be concluded from the width of the abrasion marks alone. Although the HCPEB-treated substrates in Figures 6 and 7 (and Figures 7 and 8 in the new manuscript) have wider scratches than the original samples, the wear pattern shows that the wear marks are flatter and shallower. In order to better illustrate my conclusion, I made up a graph of the wear rate data. The graph shows that the wear rate decreases as the number of pulses increases, which is corroborated by the wear profile and the friction coefficient graph.
Reviewer 3 Report
I have annotated my comments.

Author Response
Response letter
Dear Reviewer:
Thank you for your letter regarding our manuscript entitled “Effect of High-Current Pulsed Electron Beam and CeO2 Synergy on the Wear Resistance of Al-20SiC Composites” comments very carefully, and we found the comments and suggestions of the reviewers were extremely helpful.
we have carefully addressed the issues raised by the reviewer, point by point.The revised part is marked in red in the revised manuscript, We believe your comments and suggestions have significantly improved the quality of manuscript. Thanks again for your consideration of our manuscript for publication in this journal. We look forward to your favorable decision.
(1)Pls rephrase the title for better understanding
Thank you for your valuable comments. According to your comments, I have revised the title accordingly, and the revised title is;Effect of High Current Pulsed Electron Beam (HCPEB) on the organization and wear resistance of CeO2-modified Al-20SiC composites
(2)Why specifically 0.3 % used? Is it trail & error? Any Optimisation technique employed?
Thank you for your valuable comments. The parameter of 0.3% addition of rare earth CeO2 content was determined based on the research papers of related scholars and the results of the previous research of our group. The following literature can also prove that the mass fraction of CeO2 addition content at 0.3% has a better effect on Al wear resistance enhancement.
https://doi.org/10.1016/j.matlet.2022.133244
https://doi.org/10.1007/s10853-019-03892-z
(3)What %?
Thank you for your valuable comments. According to literature studies, the energy utilization during the treatment of materials with intense current pulsed electron beams is around 90%. The energy efficiency of a strong current pulsed laser beam is a relative concept and the exact value depends on several factors, such as the wavelength of the laser, pulse width, repetition frequency, power density, reflection and absorption of the optical elements, as well as specific experimental conditions and application scenarios. In general, the energy efficiency of a strong current pulsed laser beam is usually in the range of a few tens to a few percent
[1]邹建新. 强流脉冲电子束材料表面改性基础研究:在金属及金属间化合物上的应用[D].大连理工大学,2007.
https://kns.cnki.net/kcms2/article/abstract?v=3uoqIhG8C447WN1SO36whBaOoOkzJ23ELn_-3AAgJ5enmUaXDTPHrF0i2roPlj_BVUzz1IQZ5Tyur8rT59eUu_yE_oml556j&uniplatform=NZKPT&src=copy
(4)Al evavaporation possible due to low boiling point?
Thank you for your valuable comments. During the treatment of aluminum matrix composites with high current pulsed electron beam (HCPEB), aluminum may evaporate to some extent. When the high energy electron beam bombards the surface of aluminum matrix composites at temperatures that can reach 1000-4000 K, the surface layer of aluminum is heated and some of the aluminum atoms may evaporate into a gaseous state.
(5)Give example
Thank you for your valuable comments. Hu et al studied the elimination of microcracks on the surface of Al alloy samples by rare earth elements, and the results showed that the addition of trace rare earth elements reduced the density of microcracks to some extent, which led to grain refinement and enhanced the alloy's resistance in corrosion solution.
https://doi.org/10.1016/j.apsusc.2015.12.192
(6)No motivation for employing rare earth oxide CeO2, also thier unique characteristics for improving wear resistance is lacking
Thank you for your valuable comments. In response to your question, I have revised the introduction accordingly
(7)What was the substrate?
Thank you for your valuable comments. In the tube furnace sintering process, the raw billets are held in corundum arks made of alumina
(8)Show the schematic of the treatment procedure for better understanding
Thank you for your valuable comments. I am sorry that you are not able to understand the HCPEB irradiation process, but the process of intense current pulsed electron beam irradiation has been well documented and I list it below. If you think I need to add a flow chart of the irradiation process for better understanding, I can add it to the latest manuscript.
https://doi.org/10.1016/j.jallcom.2021.160651
https://doi.org/10.1016/j.surfcoat.2021.127499
https://doi.org/10.1016/j.surfcoat.2021.127796
(9)what is the mechanism?
Thank you for your valuable comments. HCPEB modification of aluminum matrix composites can cause the mechanism of meritocratic orientation of Al (111) grain surfaces mainly due to the action of the electron beam, especially the high energy and density of the electron beam. Under the electron beam bombardment, the grains in the aluminum matrix composites are rearranged and recrystallized, resulting in the formation of new grains and grain boundaries. Among them, the meritocratic orientation of Al (111) grain planes is mainly due to the following mechanisms:
Energy minimum principle: Under electron beam bombardment, grain rearrangement occurs to find the minimum energy state. For Al (111) crystal faces, the surface energy is low, so that the formation of new grain boundaries is prone to meritocratic orientation.
Principle of grain boundary energy minimization: The grain boundary energy is the connection energy between different grains inside the crystal, and is also an important factor for the stability of grain boundaries. The rearrangement of grains under electron beam bombardment may lead to the change of grain boundary energy and thus the formation of new grain boundaries. For Al (111) grain planes, the grain boundary energy is low, so that the formation of new grain boundaries is prone to meritocratic orientation.
Grain orientation effect: Grain orientation effect refers to the effect of grain orientation on material properties. Under electron beam bombardment, the new grains formed will have an effect on the material properties due to the rearrangement of the grains. For Al (111) grain planes, the orientation has a greater effect on the material properties, so the formation of new grain boundaries is prone to a meritocratic orientation.
(10)mention the temperature encountered during HCPEB ? Also SiC bioling temperature?
Thank you for your valuable comments. During HCPEB treatment, the temperature of the material can reach very high levels, usually in the range of several thousand degrees Celsius. The exact temperature depends on various factors, such as the energy and duration of the electron beam pulse, the material being treated and the experimental conditions.
SiC has a very high boiling point, estimated to be about 3,590°C at standard atmospheric pressure. However, SiC can sublimate at high temperatures, which means that it can transition directly from solid to gas without going through a liquid phase. In general, the eruption temperature of silicon carbide is higher than its melting temperature because at high temperatures, silicon carbide can be converted to the gaseous state by the sublimation process without first melting. The melting temperature of silicon carbide is about 2730°C, while its eruption temperature is generally above 3000°C.
(11)show the std. deviation for hardness values
Thank you for your valuable comments. Based on your comments, Figure 4 has been revised
(12)mention the science for improvement
Thank you for your valuable comments. Based on the comments you made, I have revised the conclusion section.

Reviewer 4 Report
I have read the manuscript titled "Effect of High-Current Pulsed Electron Beam and CeO2 Synergy on the Wear Resistance of Al-20SiC Composites" for possible publication in the journal materials.
Abstract - Needs modification. The authors should bring a summary of the research conducted.
Introduction - This requires extensive revision; the authors should bring the difference between the present work and the previous works conducted by several authors. What is the novelty of the work? It is not clear from the introduction. The authors could cite the following paper related to Al-Si alloy and composites. The authors can compare the work with other reinforcements if available.
Ganesh, M.R.S., Reghunath, N., J.Levin, M. et al. Strontium in Al–Si–Mg Alloy: A Review. Met. Mater. Int. 28, 1–40 (2022). https://doi.org/10.1007/s12540-021-01054-y
Ravinath, H., Ahammed I, I., P, H., Devan S, A., Senan V R, A., Shankar, K. V., & S, N. (2023). Impact of aging temperature on the metallurgical and dry sliding wear behaviour of LM25 / Al2O3 metal matrix composite for potential automotive application. International Journal of Lightweight Materials and Manufacture, 6(3), 416-433. doi:10.1016/j.ijlmm.2023.01.002
Materials and methods - Is this a hybrid composite? The authors are requested to show the SEM image of particles and the average size of the same. Why did the authors select pure Al rather than the alloy? What is the application of normal Aluminium?
Results - Why did the SIC evaporate as the pulse current was increased?
Could you provide me with the EDS analysis of the fabricated composite with CeO2 particles? The discussion part is very poor
In wear analysis, the authors are requested to mark the necessary characteristics on the set image
Density and porosity should be measured for the composite.
was there a homogeneous distribution of the particles in the matrix?
Discussion for the friction part needs to be strengthened.
How many trials were conducted for measuring hardness? Error bar should be mentioned
Why was the wear rate not measured? The authors have discussed wear morphology. Why is that discussed? Moreover, how many trials were conducted? For the friction test, why was only one parameter considered? What is the intended application?
Could you explain the wear mechanism? I cannot understand the FESEM images.
These are my initial comments. Kindly address these.I am recommending MAJOR revison
minor correction required
Author Response
Response letter
Dear Reviewer:
Thank you for your letter regarding our manuscript entitled “Effect of High-Current Pulsed Electron Beam and CeO2 Synergy on the Wear Resistance of Al-20SiC Composites” comments very carefully, and we found the comments and suggestions of the reviewers were extremely helpful.
we have carefully addressed the issues raised by the reviewer, point by point.The revised part is marked in red in the revised manuscript, We believe your comments and suggestions have significantly improved the quality of manuscript. Thanks again for your consideration of our manuscript for publication in this journal. We look forward to your favorable decision.
(1)Abstract - Needs modification. The authors should bring a summary of the research conducted.
Thank you for your valuable comments. According to your request, the summary section has been revised accordingly
(2)Introduction - This requires extensive revision; the authors should bring the difference between the present work and the previous works conducted by several authors. What is the novelty of the work? It is not clear from the introduction. The authors could cite the following paper related to Al-Si alloy and composites. The authors can compare the work with other reinforcements if available.
Based on your comments, we have revised the introduction. The novelty of the article lies in the first study of the change in wear resistance of Al-20SiC before and after HCPEB modification and the study of its wear resistance mechanism. We have appropriately cited the literature given by you in the introduction section. (Cited in Refs. 8 and 17, respectively)
(3)Materials and methods - Is this a hybrid composite? The authors are requested to show the SEM image of particles and the average size of the same. Why did the authors select pure Al rather than the alloy? What is the application of normal Aluminium?
Thank you for your valuable comments. In this study, the aluminum matrix composites were prepared by powder metallurgy method. Sorry for not having the SEM images of the particles before, the corresponding SEM images have been added to the new manuscript in the position of Figure 1. The corresponding dimensions of the particles are added to Table I.
Pure aluminum was chosen as the subject of this study rather than the alloy to better investigate the effect of HCPEB and CeO2 synergy on the wear resistance of Al-20SiC composites. Pure aluminum has a relatively simple composition and structure, which helps to investigate more clearly the effects of HCPEB treatment and CeO2 addition on the properties of the aluminum matrix.
Common aluminum is widely used in many fields due to its good electrical and thermal conductivity and good workability. It is often used as structural materials, electronic devices, aerospace components, food packaging, etc. Characteristics of pure aluminum in engineering applications include light weight, good corrosion resistance, weldability, and good thermal conductivity.
It is important to note that the choice of pure aluminum in the article does not mean that the alloy is not valuable or important in practical applications. Pure aluminum was chosen for the study to gain a deeper understanding of the effects of HCPEB treatment and CeO2 addition on the properties of aluminum matrix materials, which is a guideline for further optimization of alloy design and preparation processes.
(4)Results - Why did the SIC evaporate as the pulse current was increased?
Thank you for your valuable comments. During the HCPEB process, the temperature of the material can reach very high levels, usually between several thousand degrees Celsius. At high temperatures, silicon carbide can be converted to a gaseous state through a sublimation process without first melting. The melting temperature of silicon carbide is about 2730°C, while its eruption temperature is typically above 3000°C. By increasing the number of pulses, heat accumulates in the heat affected zone and finally reaches the SiC eruption temperature.
(5)Could you provide me with the EDS analysis of the fabricated composite with CeO2 particles? The discussion part is very poor
Thank you for your valuable comments. Based on your suggestion, the corresponding analysis of the EDS results has been added. Specific analysis is added to the article as well as the following
Fig. 4(e)shows that the surface of the aluminum matrix is covered with an oxide film. The oxide layer improves the wear resistance of the aluminum matrix composite surface be-cause the oxide layer has a high hardness and resists friction and wear. In addition, the oxide layer provides lubrication to the material surface and reduces the coefficient of friction.
(6)In wear analysis, the authors are requested to mark the necessary characteristics on the set image
Thank you for your valuable comments. According to your suggestion, the wear morphology SEM image has been marked accordingly
(7)Density and porosity should be measured for the composite.
Thank you for your valuable comments. According to your suggestion, the density of both samples was measured, and after several measurements the average density of Al-20SiC composite was 2.59 g/cm3, and the average density of Al-20SiC-0.3CeO2 composite was 2.64 g/cm3. The porosity of the composites was measured according to Archimedes method, and the average porosity of Al-20SiC composite was at 1.691% , and the average porosity of Al-20SiC-0.3CeO2 composites was 1.435%. It can be seen that the addition of rare earth CeO2 contributes to the density of the composite and confirms that casting defects such as porosity brought about by powder metallurgy can be eliminated. (I will add this analysis to the article if needed.)
(8)was there a homogeneous distribution of the particles in the matrix?
Thank you for your valuable comments. The powder has been well mixed by ball milling for two hours. According to the SEM surface morphology, it can be seen that SiC and CeO2 are more uniformly distributed in the aluminum matrix after HCPEB treatment
(9)Discussion for the friction part needs to be strengthened.
Thank you for your valuable comments. Thank you for your question, I have added a discussion of the wear mechanism in the last paragraph of the text.
(10)How many trials were conducted for measuring hardness? Error bar should be mentioned
Thank you for your valuable comments. All data in the text are derived from multiple measurements on different samples. The hardness data was obtained from three sets of samples, each of which was measured seven times. I am sorry that the original graph does not show the error bars clearly, I have modified the hardness data graph accordingly.
(11)Why was the wear rate not measured? The authors have discussed wear morphology. Why is that discussed? Moreover, how many trials were conducted? For the friction test, why was only one parameter considered? What is the intended application?
Could you explain the wear mechanism? I cannot understand the FESEM images.
Thank you for your valuable comments. We are sorry that the wear rate test was not performed before, we have made up the wear rate test and the corresponding data is in Figure 7. When studying the wear resistance of a material, understanding the wear morphology can help us to understand the wear mechanism of the material more deeply and thus suggest a more reasonable improvement plan. Therefore, it is very necessary to discuss the wear morphology. Each group of experiments was conducted for 5 times and more. I am sorry that only one parameter, the coefficient of friction, was considered before, so I made up the wear rate measurement. This study improves the wear resistance of the substrate, reduces the wear rate of the component, improves the strength and hardness of the material, and extends the service life of the component. And it is applied to aerospace engines, aerospace structural parts, automobile engines, automobile chassis, etc.
I have listed the corresponding wear mechanisms below and in the latest manuscript. (The exact location is in the upper paragraph of Figure 9)
The HCPEB treatment plays a pivotal role in optimizing the microstructure of Al-20SiC composites. The intense current pulses generated by HCPEB initiate rapid melting and solidification processes, resulting in improved grain orientation. This optimized microstructure significantly enhances the material's resistance to wear by minimizing crack propagation, reducing surface deformation, and mitigating fatigue. Additionally, CeO2 strengthens the bonding between the reinforcing phase (SiC) and the aluminum matrix, effectively reducing the occurrence of defects like pores and microcracks. This improved bonding enhances the overall integrity and strength of the composite material, resulting in enhanced wear resistance. Finally, CeO2 enhances the mobility of alloying elements within the material. This increased mobility allows for better dispersion of hard phase particles, such as SiC, throughout the aluminum matrix. The uniform dispersion of these hard phase particles leads to heightened hardness and improved resistance to wear and abrasion.

Round 2
Reviewer 1 Report
The revised manuscript well reflects the reviewers' comments.
Reviewer 4 Report
I appreciate the authors in revising the paper as per comments. Recommend to accept the paper
Minoe correction required